# Game Theoretical Analysis of a Multi-MNO MVNO Business Model in 5G Networks

**Erwin Jairo Sacoto Cabrera** [1,2,*] **, Luis Guijarro** [1] **and Patrick Maillé** [3]

[1]  Departamento de Comunicaciones, Universitat Politècnica de València, 46022 València, Spain; lguijar@dcom.upv.es
[2]  GIHP4C, Universidad Politécnica Salesiana-Sede Cuenca, Calle Vieja 12-30 y Elia Liut, Cuenca 010105, Ecuador
[3]  IMT Atlantique/IRISA, 35576 Rennes, France; patrick.maille@imt.fr
[*]  Correspondence: esacoto@ups.edu.ec

**Abstract:** This paper analyzes the economic feasibility of a business model for multi-Mobile Network Operators (MNOs) and Mobile Virtual Network Operators (MVNOs), which is an envisioned scenario in mobile telecommunications markets supported by 5G networks. A business model for the provision of service to end-users through an MVNO using the infrastructure support of two MNOs is proposed. We analyze the proposal though a model that captures both system and economic features. As regards the systems features, an MVNO provides service to final users using the infrastructure support of two MNOs. The agreement between MVNO and MNOs is such that the MVNO will split the network traffic between the two MNOs and will pay to each MNO for the traffic served through its infrastructure. As regards the economic features, the incentives are modelled through the user utilities and the operators' profits; and game theory is used to model the strategic interaction between the users' subscription decision and the MNO network capacities decision. We conclude that such a model is feasible from an economic point of view for all the actors.

**Keywords:** mobile network operators; 5G; mobile virtual network operator; game theory; nash equilibrium; queue theory

---

## 1. Introduction

In recent years, we have witnessed the rapid growth of the Mobile Telecommunications Market, thanks to the strong implementation of fourth-generation (4G) networks and the development of fifth-generation (5G) networks [1,2]. This has enabled the introduction of new business models and innovative types of services [3]. Furthermore, the architecture flexibility and scalability of those networks allow sharing any sort of network resource between Mobile Network Operators (MNOs) and Mobile Virtual Network Operators (MVNOs). This supports a new generation of MVNOs as described in [4–6]. Besides, the evolution of mobile networks has reduced the barriers to MVNO entry and facilitated MNO–MVNO partnerships [7]. As a result, global MVNO market revenue reached about $60.5 billion in 2018 [8] and is forecast to grow to $103 billion in 2023 [9].

Currently, this new multi-market and multi-technological environment has allowed MVNOs to partner with multiple MNOs and operate across their networks [7,10]. This MNO–MVNO partnership can potentially expand the MNO market share and bring new revenue [10]. However, in order to offer a service, an MVNO needs to lease the resource under a specific sharing agreement with one or more MNOs to utilize their network resources [11]. Such a sharing agreement requires MNOs to consider the impact on their revenue by enabling the MVNO to select the proportion of network resources that it will contract with each MNO. The key elements to convince MNOs regarding entry with an MVNO

is the access fee for MNO resource use and limiting the limitation of MVNO's priority to access MNOs' infrastructure [12].

Nowadays, multi-MNO MVNO partnership is strongly supported by the main objectives of the engineering and architecture of 5G networks as described in [13–16]. Accurately, in [13,14], the authors describe 5G network framework and introduce to new business models, such as Network Sharing, Connectivity Providers and Partner Service Providers. In [15,16], the authors introduce the concept of 5G Network Slice Broker in 5G systems, which enables mobile virtual network operators, over-the-top providers, and industry vertical market players to request and lease resources from infrastructure providers dynamically via signalling means. Specifically, in the context of network slicing (NS), in recent studies on 5G state-of-the-art related to the principles, characteristics and new business models based on NS [17–22], conclude that the NS provides an optimal solution for multiple scenarios which demands specific requirements for operators and users not foreseen in 4G networks. Besides, in the mentioned studies propose several issues for future research, which we highlight the following: New network services, multi-operator business models, dynamic slicing, flexibility in allocation capacity, users' quality of service (QoS), and pricing services. Furthermore, in [23], the authors analyze the technical characteristics to implement a high-level Virtual Operator (multi-MNOs and MVNO partnership) as a network sharing model business. The above study has shown that 5G network technologies allow high-level virtual operators to solve the problem of building their network and bringing 5G services to the market with added value using these technological capabilities. Finally, realistic multi-MNO business model cases are described in [24] where companies (MVNOs) that resell connectivity to their customers could bring slices of different carriers (MNOs) together to offer a service with specific characteristics.

In the above scenario, NS is gaining increasing importance as an effective way to introduce flexibility in network resources management. Among the different use cases that NS will enable in the upcoming 5G, the focus of this work is on the analysis of the economic viability of this possible partnership between an MVNO and multiple MNOs as enabled by 5G networks; we call that relationship "multi-MNO MVNO". In this work, we study the feasibility of the model from a positive-profit point of view for all the actors. This model is analyzed as a multi-stage game to examine the equilibrium decisions of the MVNO, MNOs and users in such a setting. In particular, we characterize the conditions under which it is beneficial for MNOs to collaborate with the MVNO, taking into account the capacity settings adopted by all operators and the consequences on the respective market shares and users benefits [25].

To test the above described, we propose a business model to be implemented by two MNOs and an MVNO, and we analyze the proposal using a strategic game (Section 2.3). In this strategic game, we capture the users' utilities, MVNO traffic split and the operators' profits; and we analyze the strategic interaction among the MNOs in order to maximize their profits, the MVNO's decision to split its network traffic to MNOs and the users' decision whether or not to subscribe to the MVNO. Besides, we compared the results obtained in the multi-MNO MVNO Model with the results obtained in a model in which an MVNO shares its network traffic with only one MNO; we call this model single-MNO.

The main contributions of this paper are the following:

- A business model proposal (multi-MNO MVNO) to provide service to end-users through an MVNO using the infrastructure support of two MNOs and analyze the interactions between the different actors (MVNO, MNOs and users).
- A viability analysis of an agreement between an MVNO and MNOs, wherein the MVNO will distribute the users' traffic between to MNOs and will pay to each MNO for the traffic served through its infrastructure.
- A thorough mathematical analysis of the Nash Equilibria for the game played by the MVNO and both MNOs is carrying out.

In the analysis, we apply microeconomics concepts and queuing theory in the formulation of the multi-MNO MVNO association model and the modelling of user-perceived QoS. Game theory is used in the paper to analyze the competition among MNOs. Game Theory is a branch of economic theory that aims to help understand the interactions among decision-makers [26,27]. It is widely used in telecommunications and computer network system models in order to optimize routing (see, e.g., the work on networks optimization and control by Srikant et al. [28], and the work on designing routing protocols in Wireless Sensor Networks (WSNs) by Habib et al. [29]), resource sharing (see, e.g., the work on models of resource allocation algorithms in 5G NS by Ruoyu Su et al. [30], and the work on resource management aspects of NS where recent solutions are discussed and compared by et al. [18]), and the economic incentives of the agents, either users or providers (see, e.g., Antoniou et al.'s work [31] and the work on competition in data-based service provision by Guijarro et al. [32]). Our work focuses on the issues of multi-operator, game theory, and optimization schemes which are part of NS taxonomy described in [20]. Besides, our work contributes to the key points and main open issues related to NS research [13,14], therefore, it belongs to a latter trend, and it shares this feature with some of the works referred to in the next subsection.

*Related Work*

This work is inspired by the business model descriptions presented in [4] which can be classified as: "full MVNO" where virtual operators install their own core equipment, " multi-MNO" model where the MVNO connects to several network providers, and "Always Best Connected (ABC)" model allows an MVNO to choose the best MNO for each connection. In this context, some novel situations have come up with the MNO–MVNO relationship, that has been addressed in several studies.

Specifically, many recent works consider an MVNO being the client of their host MNO but also competing with MNOs and other MVNOs to attract customers. For example, the authors in [33] analyze the light MVNO model and the wholesale discount offered by the MNO on the fulfilment of certain conditions set out in an agreement for the resale of the mobile service. References [6,9,15,34,35] analyze the MNO multi-tenancy approaches, that are increasingly gaining momentum, paving the way toward further decreasing capital expenditure and operational expenditure (CAPEX/OPEX) costs. These analyzes tackle the context of the 5G networks. For example, the authors in [36] argued that the MVNO business can be profitable at an initial stage when it is associated with MNOs with a small market share. The authors in [27] analyze stable markets and conclude that the MVNO is better off when it preferably partners with a big MNO. The above studies have shown technical and economic feasibility of the partnership between one or several MVNOs and a single MNO, whereas, in our article, we analyze the relationship between multiple MNOs and an MVNO from an economic point of view.

There are only a few works, however, that model the economic relationships that emerge of a multi-MNO MVNO business model. A price analysis in a mobile market driven by two MNOs and a new competitor MVNO through a game-theoretical approach which is performed in [7]. The MNO investment, the MVNO decision on the MNO lease and the retail price are analyzed in [34]. Our work differs from those mentioned above because our scenario incorporates: (i) The users decision to subscribe to the MVNO based on the QoS and the price of the service, (ii) the MVNO choosing the optimal network traffic to be assigned to each MNO, and (iii) the costs incurred by the operators.

Within the scope of the MVNO business analysis, some research complements the study of the economic feasibility of different network sharing scenarios, queuing theory concepts, as well as microeconomics and game theory concepts. For instance, in the context of network sharing, in [37], the authors analyze a NS scenario, based on game theory and a framework called "share-constrained proportional allocation"; the results obtained provide an effective and implementable scheme for dynamically sharing resources across slices. An economic analysis for allocating NS requests, considering the maximum cost benefit from a MNOs perspective based on optimal stopping theory, is elaborated in [38]. In [39], the authors provide a discussion of two different dimensions for the operator

profit modelling on the concept and system architecture of NS with particular focus on its business aspect and mathematical profit modelling. In the same way, some studies analyze network sharing, based on NS, such as [40–43]. The above studies have shown the technical feasibility of the partnership between one or several MVNOs and a single MNO, whereas, in our article, we analyze the relationship between multiple MNOs and an MVNO.

In the framework of the queue theory, in [44], the effects of queuing delays and related costs on the management and control computing resources are analyzed. An analysis from a pricing perspective, based on priority queuing (PQ) and Generalized Processor Sharing, with the issue of maximizing network operator revenue, is carried out in [45]. The cooperation strategies among mobile network operators competitors, customers, and different types of partners based on network sharing have been studied in [46] using queue theory. In [47], the authors investigate the priority queuing as a way to establish service differentiation; to do that, they consider the Discriminatory Processor Sharing discipline for two models of service with different QoS and determine the prices that maximize the provider's profit. Similarly, some studies represent the functions of users demand through M/M/m queues, such as [48–50]. The above studies are shown that users' benefits are represented by an association between price and QoS, as outlined in our study.

In the framework of the game theory, in [51], for solving the spectrum sharing problem in cognitive radio networks, Nash equilibrium was considered as the solution. The authors in [52], analyzed a model where two MVNOs compete for the users in terms of QoS, by strategically distributing its share of the aggregated cells capacity managed by the InP among its subscribers; to solve that, they use the Nash equilibria. In [53,54], the authors analyze the profit maximization problem of a set of independent MVNOs that request slices from an MNO, and propose the game theory to solve several allocation mechanisms for solving those optimization problems. In the same way, in some studies, game theory to solve different scenarios is analysed. Works apply game theory to the analysis of service provision in a competitive environment, within the context of telecommunications, such as [49,50,55]. However, our work differs importantly from these works in that we model the relationship between two MNOs, an MVNOs and the users as different agents with their particular incentives, which are the profits and the user utility, respectively.

Finally, our paper relates to work [56] that it uses a multi-stage game to analyze the equilibrium decisions of the MVNO, MNOs, and users in such a setting. That work characterizes the price competition as a Stackelberg game in three-stages, where the MVNO can collaborate with the MNOs under certain spectrum sharing conditions. The above study used game theory to show that the MVNO can compete with MNOs in the same market for prices, whereas, in our article, we use game theory to model the strategic interaction between users' subscription and the MNOs network capacities decision. In addition, our analysis generalizes the applicability of the model multi-MNO MVNO beyond a specific resource sharing as analyzed in [56], where we provide a more detailed analytical derivation. Specifically, we propose a business model for an MVNO, where the MVNO provides service to final users using the infrastructure support of two MNOs. The agreement between the MVNO and the MNOs is such that the MVNO will split the network traffic between the two MNOs and will pay to each MNO for the traffic served through its infrastructure. Moreover, this business model is proposed by flexibility foreseen in the 5G networks architecture [23]. Our objective is to search for an equilibrium, given that every stakeholder (users, MVNO, MNOs) has its own preferences.

The remainder of this paper is organized as follows. In Section 2, the multi-MNO MVNO model is described. In Section 3, an analysis of the MVNO service provision, network traffic split decision and MNOs network capacity strategies is presented. In Section 4, the numerical results are discussed to illustrate the analysis. Finally, Section 5 draws the conclusions.

## 2. Model Description

In this section, we consider a scenario in which the final users are served by an MVNO and its network traffic is split between two MNOs that provide the infrastructure support to the MVNO.

The scenario modelled in this work is illustrated in Figure 1 from two perspectives: The system perspective and the economic perspective. In short:

- the MVNO provides service to final users;
- MNOs carry out that service for the MVNO;
- final users will determine if they subscribe or not with the MVNO. Moreover, operators profits (MVNO and MNOs) depend on the users' subscription decisions to MVNO.

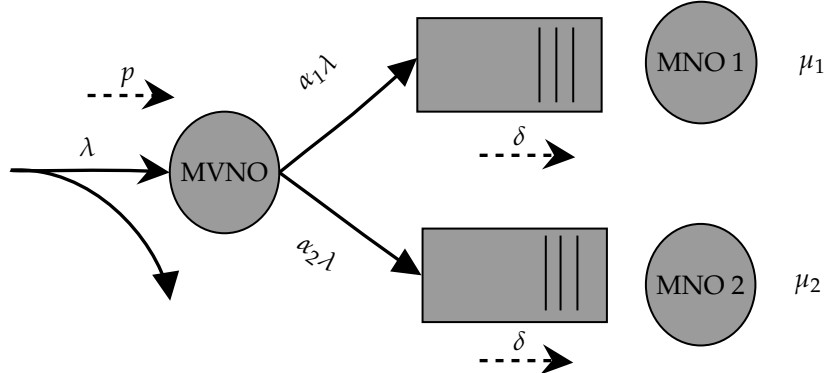

**Figure 1.** System Model: Users mean arrival rate ($\lambda$), Mobile Virtual Network Operator (MVNO) split factor ($\alpha_i$), Mobile Network Operator (MNO)$_i$ packets mean arrival ($\lambda_i$), MNO $i$ network capacity ($\mu_i$), price charged MVNO ($p$), fee paid by the MVNO ($\delta$), network traffic flow ($\longrightarrow$), price and fees flow ($--\rightarrow$).

Mobile Network Operator (MNO) and Mobile Virtual Network Operators (MVNOs).

Besides, in Appendix A we analyze the case where there is only one MNO providing service; i.e., the MVNO will send the users traffic only to one monopolistic MNO and will pay it for the traffic served. That case will be used as a baseline for comparison in Section 4 to establish whether model multi-MNO MVNO presents better conditions for all the actors (operators and users). In the following subsections, we describe the model in detail. A summary of the notation used in this paper is given in Table 1.

**Table 1.** Summary of notation.

|  |  | Equation |
|---|---|---|
| **General Model** |  |  |
| MNO $i$'s mean packet system time | $T_i$ | (1) |
| MNO $i$'s network capacity | $\mu_i$ | (1) |
| MNO $i$'s packets mean arrival rate | $\lambda_i$ | (1) |
| MVNO packets mean arrival rate | $\lambda$ | (2) |
| MVNO traffic split factor | $\alpha_i$ | (2) |
| MVNO mean packet system time | $T$ | (1) |
| Quality perceived by the users | $Q$ | (6) |
| Conversion factor from $s^{-1}$ to monetary units | $c$ | (6) |
| Users utility | $u_m$ | (7) |
| **Economic Model** |  |  |
| Price charged by MNO | $p$ | (7) |
| Fee paid by the MVNO | $\delta$ | (7) |
| MVNO profits | $\Pi_m$ | (9) |
| MNO profits | $\Pi_i$ | (10) |
| Constant of the unit cost of acquisition | $K_i$ | (10) |
| Cost adjustment parameter | $q$ | (10) |

**Table 1.** *Cont.*

|  |  | Equation |
|---|---|---|
| **Analysis** |  |  |
| MVNO optimal packets mean arrival | $\lambda^*$ | (12) |
| MVNO optimal profits | $\Pi_m^*$ | (15) |
| MVNO optimal traffic split factor | $\alpha_1^*$ | (2) |
| MNO 1's best response | $BR_1$ | (21) |
| MNO 2's best response | $BR_2$ | (22) |
| MNO 1's equilibrium capacity | $\mu_1^*$ | (23) |
| MNO 2's equilibrium capacity | $\mu_2^*$ | (24) |
| **Appendix A** |  |  |
| MNO 0's profits—single-MNO model | $\Pi_0$ | (A1) |
| Constant of the unit cost of acquisition—single-MNO model | $K_0$ | (A1) |
| Cost adjustment parameter—single-MNO model | $q_0$ | (A1) |
| Users Utility | $U_{m_0}$ | (A4) |
| MVNO profits—single-MNO model | $\Pi_{m_0}$ | (A4) |

*2.1. System Model*

We model the service infrastructure of an MNO as an M/M/1/∞ queue [57]. To model a whole network as a single M/M/1 queue is a simplification justified by the need to obtain manageable expressions for the utility of the network users. This approach has been taken previously by [45,47,58] in the context of economic analysis of the Internet service. The users generate information packets that feed into the system at rate $\lambda$. In our model, we use a single server queue (MVNO) that does not make any distinction among the packets in a FIFO queue. We assume that the service times of the individual packets are i.i.d. exponentially distributed random variables, with mean $\mu_i^{-1}$, where $\mu_i$ can be interpreted as the MNO $i$'s network capacity.

The relevant QoS metric is the mean packet system time, which comprises both the waiting time and the service time. The MNO's mean packet system time $T_i$ is given by

$$T_i = \frac{1}{\mu_i - \lambda_i}, \quad i = 1, 2, \tag{1}$$

$\lambda_i$ is the expected value of the arrival rate of users distributed by Poisson to MNOs networks.

The MVNO partitions the stream in two traffic flows so that a Poisson-distributed flow with mean arrival rate $\lambda_1 = \alpha_1 \lambda$ is forwarded to MNO 1, and a Poisson-distributed flow with users mean arrival rate $\lambda_2 = \alpha_2 \lambda$ is forwarded to MNO 2. We assume for stability reasons that $\lambda_i < \mu_i$, i.e., for each flow from the MVNO to the MNOs we will have

$$\alpha_1 \lambda < \mu_1, \tag{2}$$

$$\alpha_2 \lambda < \mu_2, \tag{3}$$

where $\alpha_i$ is the MVNO traffic split factor and $\lambda$ is the MVNO packets mean arrival rate.

If the system operates as described above, the aggregated packet flow bears an average system delay through the parallel combination of both MNO's network given by

$$T = \alpha_1 T_1 + \alpha_2 T_2, \tag{4}$$

where $\alpha_2 = 1 - \alpha_1$, and substituting Equations (1) in (4) we obtain

$$T = \frac{\alpha_1}{\mu_1 - \alpha_1 \lambda} + \frac{1 - \alpha_1}{\mu_2 - (1 - \alpha_1)\lambda}. \tag{5}$$

The valuation for the service perceived by each user is proposed to be given by the expression used in [47,58,59]

$$Q = c \ T^{-1}, \tag{6}$$

where $c > 0$ is a conversion factor to express the valuation in monetary units. Note that $Q$ decreases with $T$, which means that the higher the delay, the worse the quality.

## 2.2. Economic Model

The user utility $u_m$ is modelled as the difference between their valuation and the price charged by the operator.

$$u_m = Q - p, \tag{7}$$

with $p$ the price charged by the MVNO, which gives from Equation (5)

$$u_m = \frac{c}{\frac{\alpha_1}{\mu_1 - \alpha_1 \lambda} + \frac{1 - \alpha_1}{\mu_2 - (1 - \alpha_1)\lambda}} - p. \tag{8}$$

The utility of the alternative option, i.e., no subscribing, is set to zero, i.e., $u_0 = 0$. Note that a similar approach for modeling the user utility can be found in, e.g., [12,47,48,58,59]. Moreover, this form of utility function can be related to the quasi-linear function widely used in microeconomic and telecommunications networks analysis [27]. The user utility has the advantage of being interpretable in monetary-equivalent terms, i.e., allows to relate the QoS perceived by the users and the paid price for the service.

The MVNO obtains revenue from the users equal to $\lambda p$, but it must pay a fee $\delta$ (m.u.) for each packet serviced by the MNOs. The MVNO's profits are then given by

$$\Pi_m = \lambda p - \alpha_1 \lambda \delta - (1 - \alpha_1)\lambda \delta = (p - \delta)\lambda. \tag{9}$$

Finally, MNO $i$ gets a revenue per user equal to $\delta$ and bears an investment cost related to the capacity equal to $K\mu_i + q\mu_i^2$ so that its profit is given by

$$\Pi_i = \alpha_i \lambda \delta - K_i \mu_i - q\mu_i^2. \quad i = 1, 2. \tag{10}$$

We assume that investment cost $(K_i \mu_i + q\mu_i^2)$ to be a strictly convex function of the rate of investment as defined in [60], parameterized by $k \geq 0$ and $q > 0$.

## 2.3. Strategic Game

We assume that the MVNO and each MNO have their own incentives and make decisions in order to maximize their respective profits. We assume that prices are fixed and therefore all stakeholders are price-takers. This assumption can be justified by a scenario where the regulator fixes both retail and wholesale prices or a situation where contracts are agreed for a time period longer than the interval where the following interaction takes place.

First, each MNO decides how much to invest by choosing a network capacity value $\mu_i$, in order to maximize its profit $\Pi_i$. Each MNO's decision influences the competitor's profit by impacting $\lambda$ and $\alpha_i$. Or in other words, the competition in network capacities among the MNOs is a simultaneous-move strategic game.

Second, given networks capacities $\mu_1$ and $\mu_2$, the MVNO decides how to split the network traffic by choosing $\alpha_1$.

Third, given network capacities $(\mu_1, \mu_2)$ and split factors $(\alpha_1, \alpha_2)$, users must decide whether to subscribe (obtaining utility $u_m$) or not (obtaining utility $u_0$). There is a strategic interaction between each individual user decision, through the congestion effect in the utility $u_m$. Under the assumption, the equilibrium reached is the one postulated by Wardrop [61] in 1952 as a rule to

solve the traffic assignment problem, i.e., a problem that concerns the selection of routes between origins and destinations in transportation networks. Specifically, Wardrop's first principle is the relevant one, which says that: The journey times on all routes actually used are equal, and less than those which would be experienced by a single vehicle on any unused route. Basically, at a Wardrop equilibrium, the utility that every user gets is equalized between the alternative effectively chosen by the users. That means that, as regards Operator $i$'s user base, either (1) $u_m = 0$ (i.e., subscription and no-subscription utilities are equal) and some users subscribe and some other users do not ($\lambda^* \geq 0$), or (2) $u_m < 0$ and no user subscribes ($\lambda^* = 0$; i.e., no user chooses the option with less than zero utility). Note that the third alternative ($u_m > 0$) cannot be sustained under Equations (1) and (7), since an increase in $\lambda$ causes an increase in $T_i$ and will eventually settle in $u_m = 0$. As described above, whereby the fraction of users subscribing service $\lambda$ will be such that

$$u_m = u_0. \tag{11}$$

The decision of the MVNO is taken with the knowledge of each MNO's decision, and the users' decision is taken with the knowledge of the MVNO's decision. The strategic interaction between the MNOs, the MVNO and the users is of a sequential-move game [27,62]. Furthermore, the structure of this game is a three-stage game as depicted in Figure 2. A standard way to analyze this sort of games is by means of backward induction: This method proceeds in the opposite direction to the one in which decisions are taken. Firstly, we consider the user subscription decision, i.e., each user takes its own subscription decision, trying to maximize the utility it gets from either subscribing to the MVNO or not. Secondly, we obtain the optimal $\alpha_1$ with which the MVNO will split the network traffic to MNOs. Thirdly, each MNO chooses a network capacity to maximize its profits simultaneously and independently. Each operator is not only aware of the MVNO decision in the second stage, but also of the rational behaviour of the other operator.

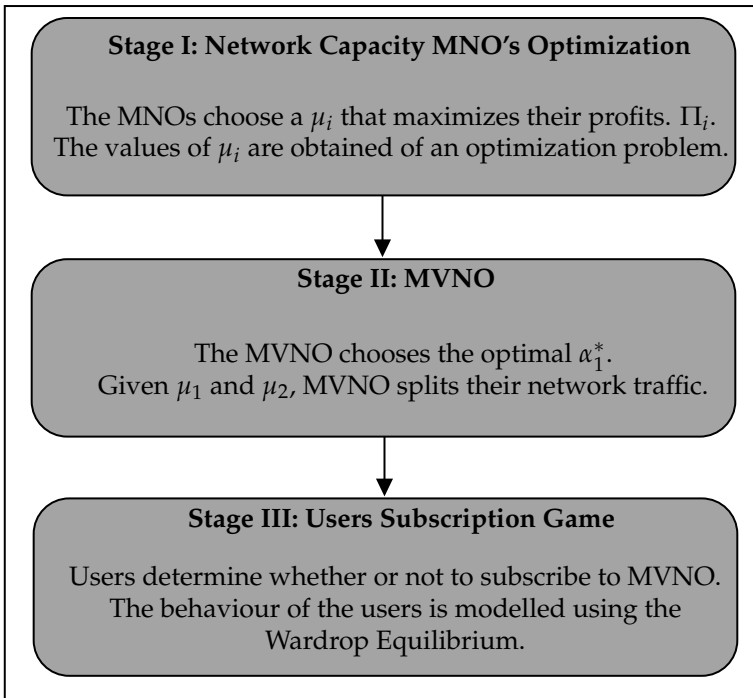

**Figure 2.** Description of the game stages.

## 3. Analysis

Our objective is to compute the Nash equilibrium of the three-stage game described above and analyze the effect of the different parameters on the equilibrium strategies and payoffs.

### 3.1. MVNO Service Provision

In this section, we study the users' subscription to the MVNO service. We assume that users are infinitesimal and therefore their choices do not individually affect the subscription levels of the MVNO. The outcome resulting from such user interactions is described by Wardrop's principle [61], i.e., a user will enter the system as soon as his utility $u_m$, is positive, or leave the system if it is negative. The following proposition characterizes the user subscription corresponding to the utility perceived by the users.

**Proposition 1.** *For any fixed values of $\mu_1 > 0$, $\mu_2 > 0$, $\alpha_1 \in [0,1]$, c, and p, there is a unique user equilibrium mean arrival rate $\lambda$, given as follows.*

- *If $\frac{\alpha_1}{\mu_1} + \frac{1-\alpha_1}{\mu_2} \geq \frac{c}{p}$ then $\lambda = 0$.*
- *Otherwise, $\lambda > 0$ is the unique solution in $\left[0, \min\left(\frac{\mu_1}{\alpha_1}, \frac{\mu_2}{1-\alpha_1}\right)\right)$ of $\frac{\alpha_1}{\mu_1 - \alpha_1 \lambda} + \frac{1-\alpha_1}{\mu_2 - (1-\alpha_1)\lambda} = \frac{c}{p}$ and equals*

$$\lambda^* = \begin{cases} \mu_2 - \dfrac{p}{c} & \text{if} \quad \alpha_1 = 0 \\[2mm] \mu_1 - \dfrac{p}{c} & \text{if} \quad \alpha_1 = 1 \\[2mm] \dfrac{\mu_1}{2\alpha_1} + \dfrac{\mu_2}{2(1-\alpha_1)} - \dfrac{p}{c} - \sqrt{\left(\dfrac{\mu_1}{2\alpha_1} - \dfrac{\mu_2}{2(1-\alpha_1)}\right)^2 + \dfrac{p^2}{c^2}}, & \text{if} \quad \alpha_1 \in (0,1) \end{cases} \tag{12}$$

**Proof of Proposition 1.** An user mean arrival rate $\lambda$ satisfies the equilibrium condition if and only if

- either $\lambda = 0$ and the utility of entering the system is non-positive (so that nobody will join), i.e.,

$$\frac{c}{\frac{\alpha_1}{\mu_1} + \frac{1-\alpha_1}{\mu_2}} - p \leq 0 \tag{13}$$

   where we plugged $\lambda = 0$ in Equation (8),
- or the utility of entering the system is zero, i.e., $\lambda$ is a solution of

$$\frac{c}{\frac{\alpha_1}{\mu_1 - \alpha_1 \lambda} + \frac{1-\alpha_1}{\mu_2 - (1-\alpha_1)\lambda}} - p = 0.$$

Condition (13) is equivalent to $\frac{1-\alpha_1}{\mu_2} + \frac{\alpha_1}{\mu_1} \geq \frac{c}{p}$, which proves the first part of the proposition.

Now assume that condition in Equation (13) is not satisfied: On the interval $[0, \min(\mu_1/\alpha_1, \mu_2/(1-\alpha_1)))$ the continuous function $\lambda \mapsto \frac{\alpha_1}{\mu_1 - \alpha_1 \lambda} + \frac{1-\alpha_1}{\mu_2 - (1-\alpha_1)\lambda} - \frac{c}{p}$ is then strictly increasing, strictly negative at 0 and tends to infinity when $\lambda$ tends to $\min(\mu_1/\alpha_1, \mu_2/(1-\alpha_1)) < \infty$, with the convention that $\mu_i/0 = \infty$ for $i = 1, 2$. Therefore, there exists a unique solution of the equation

$$\frac{\alpha_1}{\mu_1 - \alpha_1 \lambda} + \frac{1 - \alpha_1}{\mu_2 - (1 - \alpha_1)\lambda} = \frac{c}{p}, \tag{14}$$

that is the equilibrium $\lambda$ we are looking for. Moreover, that solution is strictly positive.

There remains to find that solution analytically:

- if $\alpha_1 \in \{0, 1\}$ the solution is trivial;
- otherwise, Equation (14) is equivalent to $\lambda$ being the unique root in $[0, \min(\mu_1/\alpha_1, \mu_2/(1-\alpha_1)))$ of $g(\lambda) := \alpha_1(\mu_2 - (1 - \alpha_1)\lambda) + (1 - \alpha_1)(\mu_1 - \alpha_1 \lambda) - \frac{c}{p}(\mu_1 - \alpha_1 \lambda)(\mu_2 - (1 - \alpha_1)\lambda)$, which we can rewrite as

$$g(\lambda) = -\frac{c}{p}\alpha_1(1 - \alpha_1)\lambda^2 + \left(\frac{c}{p}[\alpha_1\mu_2 + (1 - \alpha_1)\mu_1] - 2\alpha_1(1 - \alpha_1)\right)\lambda + \alpha_1\mu_2 + (1 - \alpha_1)\mu_1 - \frac{c}{p}\mu_1\mu_2,$$

a degree-two polynomial equation in $\lambda$, with at least one positive root since an equilibrium exists, from the reasoning above. Equation (13) does not hold, $g(0) < 0$. Additionally, notice that $g(\cdot)$ is strictly concave, therefore all its roots are strictly positive. However, there is only one root in $[0, \min(\mu_1/\alpha_1, \mu_2/(1 - \alpha_1)))$, hence, we use the classical expression for the smallest root of a degree-two polynomial, and a bit of algebra, to get the expression in Equation (12).

□

We note that the expression for $\lambda$ given in Equation (12) is defined for $\mu_1 > 0$, $\mu_2 > 0$ and $\alpha_1 \in [0, 1]$ (see the Proof of Proposition 1), which is a compact set, and is continuous (and differentiable) in this domain.

From this, we can obtain the MVNO profits $\Pi_m^*$ substituting Equations (12) into Equation (9).

$$\Pi_m^*(\alpha_1) = (p - \delta)\left(\frac{\mu_1}{2\alpha_1} + \frac{\mu_2}{2(1 - \alpha_1)} - \frac{p}{c} - \sqrt{\frac{\mu_1^2}{4\alpha_1^2} + \frac{p^2}{c^2} - \frac{\mu_1\mu_2}{2(1 - \alpha_1)\alpha_1} + \frac{\mu_2^2}{4(\alpha_1 - 1)^2}}\right) \quad \text{if} \quad \alpha_1 \in [0, 1]. \quad (15)$$

Note that when $\alpha_1 = 0$ or $\alpha_1 = 1$ the MVNO select a single MNO to send its network traffic. Therefore the MVNO profits are given by

$$\Pi_m^*(\alpha_1) = \mu_1(p - \delta) - \frac{p}{c}(p - \delta) \quad \text{if} \quad \alpha_1 = 0, \quad (16)$$

$$\Pi_m^*(\alpha_1) = \mu_2(p - \delta) - \frac{p}{c}(p - \delta) \quad \text{if} \quad \alpha_1 = 1. \quad (17)$$

*3.2. MVNO Decision*

In this section, we investigate how the MVNO splits its network traffic among MNOs through the parameter $\alpha_1$. Since the MVNO revenue is proportional to the amount of the users mean arrival rate $\lambda$ (see Equation (9)), the MNOs selection problem refers to finding a value of $\alpha_1$ maximizing $\lambda$ (whose expression is given in Proposition 1) as a function of the capacities offered by the MNOs. The following proposition indicates how an MVNO should choose $\alpha_1$.

**Proposition 2.** *Assume fixed values of $\mu_1 > 0$, $\mu_2 > 0$, $\delta$, $c$ and $p > \delta$.*

- *If $\max(\mu_1, \mu_2) \le \frac{p}{c}$, then $\lambda = 0$ for any $\alpha_1$ so there is no $\alpha_1$ maximizing $\lambda$ (the MVNO always get a revenue 0);*
- *Otherwise, these is a unique $\alpha_1$ maximizing the MVNO revenue, given by*

$$\alpha_1^* = \begin{cases} \dfrac{1}{2} & \text{if} \quad \mu_1 = \mu_2 \\[2ex] \dfrac{\sqrt{\left(\frac{(c(\mu_1 + \mu_2)}{p}\right)^2 - 8\frac{c}{p}\sqrt{\mu_1\mu_2} + 4 - c\frac{\mu_1 + \mu_2}{p} + 2\sqrt{\frac{\mu_1}{\mu_2}}}}{2\left(\sqrt{\frac{\mu_1}{\mu_2}} - \sqrt{\frac{\mu_2}{\mu_1}}\right)} & \text{otherwise.} \end{cases}$$

**Proof of Proposition 2.** From Equation (9), if $p > \delta$ then maximizing the MVNO revenue is equivalent to maximizing the network traffic $\lambda$.

From Proposition 1, for a given $\alpha_1$ the network traffic $\lambda$ is strictly positive if and only if $\frac{\alpha_1}{\mu_1} + \frac{1 - \alpha_1}{\mu_2} - \frac{c}{p} < 0$. Therefore,

- if $\frac{\alpha_1}{\mu_1} + \frac{1 - \alpha_1}{\mu_2} - \frac{c}{p} \ge 0$ for all $\alpha_1 \in [0, 1]$ then looking for an optimal $\alpha_1$ makes no sense since user mean arrival rate is always null. That condition is equivalent to $\min_{x \in [0,1]} \frac{x}{\mu_1} + \frac{1 - x}{\mu_2} - \frac{c}{p} \ge 0$, or $\min\left(\frac{1}{\mu_1}, \frac{1}{\mu_2}\right) \ge \frac{c}{p}$, or again $\max(\mu_1, \mu_2) \le p/c$.
- Otherwise, we know that there exists some $\alpha_1 \in [0, 1]$ eliciting a strictly positive user mean arrival rate $\lambda$; our goal is now to find an $\alpha_1$ maximizing $\lambda$. Note from Equations (12) that the network traffic is a continuously differentiable function of $\alpha_1$ in the open interval (0,1).

We can maximize the $\lambda^*(\alpha_1)$ setting its derivative with respect to the $(\alpha_1)$ equal to zero and checking if the solution obtained is a maximum:

$$\alpha_1^* = \frac{\sqrt{\left(\frac{(c(\mu_1+\mu_2))}{p}\right)^2 - 8\frac{c}{p}\sqrt{\mu_1\mu_2} + 4} - c\frac{\mu_1+\mu_2}{p} + 2\sqrt{\frac{\mu_1}{\mu_2}}}{2\left(\sqrt{\frac{\mu_1}{\mu_2}} - \sqrt{\frac{\mu_2}{\mu_1}}\right)}. \tag{18}$$

□

### 3.3. MNO's Simultaneous-Move Strategic Game

In this section, we concentrate on the MNO's network capacity $(\mu_1^*, \mu_2^*)$. It is a selection problem that aims to achieve their maximum profits. For this purpose, we consider the results of the previous stage as $\lambda^*$ and $\alpha_1^*$. From Equations (10), it can be inferred that MNOs profit depends not only on $\mu_i$. This dependence can be made explicit as follows: $\Pi(\mu_1, \mu_2)$. Moreover, this dependence is strategic, that is, MNOs takes acts independently and selfishly. Game theory provides the theoretical foundation for analyzing this strategic relationship. Specifically, since each MNO acts independently and selfishly, the appropriate game-theoretical models are the non-cooperative ones.

In our model, the players are the MNOs, the strategies are the capacities, and the incentives are the MNOs profits. Then, turning our attention to the capacities between the MNOs, the equilibrium strategies $\mu_1^*$ and $\mu_2^*$ are given by the Subgame Perfect Equilibrium (SPNE) conditions [26]. We use the SPNE as a solution concept, whereby, at the first stage, the MNO's network capacity $\mu_i$ that chooses are such that it gets no revenue improvement from changing the MNO's network capacity assuming that the competitor MNO $i$ do not deviate from the equilibrium capacities, anticipating the decision of the previous stages. To sum up, given $\lambda^*$ and $\alpha_1^*$, MNOs will choose $\mu_i^*$ so that

$$\Pi_1(\mu_1^*, \mu_2) \geq \Pi_1(\mu_1, \mu_2), \quad \forall \, \mu_1, \tag{19}$$

$$\Pi_2(\mu_1, \mu_2^*) \geq \Pi_1(\mu_1, \mu_2), \quad \forall \, \mu_2, \tag{20}$$

meaning that no operator can unilaterally increase its profits by a capacity.

The general method to discover the set of Nash equilibria is to obtain the best-response (BR) function of each operator an identify the crossing points. The $BR_i$ functions are defined as follows:

$$BR_1(\mu_2) = \underset{\mu_1}{\operatorname{argmax}} \, \Pi_1(\mu_1, \mu_2), \tag{21}$$

$$BR_2(\mu_1) = \underset{\mu_2}{\operatorname{argmax}} \, \Pi_1(\mu_1, \mu_2). \tag{22}$$

Once we have obtained the BR functions we can obtain the set of Nash equilibria solving the following system of equations

$$\mu_1^* = BR_1(\mu_2^*), \tag{23}$$

$$\mu_2^* = BR_2(\mu_1^*). \tag{24}$$

When solving the equilibrium equations for the third and second phase, $\lambda$ may be expressed as functions of $\mu_1$ and $\mu_2$, so that the operators' profits are a function of $\mu_1$ and $\mu_2$ only. In order to solve the above described, we substitute Equations (12) and (18) into Equation (10).

$$\Pi_1(\mu_1, \mu_2) = \alpha_1^* \lambda^* \delta - K_1 \mu_1 - q\mu_1^2, \tag{25}$$

$$\Pi_2(\mu_1, \mu_2) = (1 - \alpha_1^*)\lambda^* \delta - K_2 \mu_2 - q\mu_2^2. \tag{26}$$

We consider that profits $\Pi_1 > 0$ and $\Pi_2 > 0$, i.e., from Equations (25) and (26)

$$K_1 < \frac{\alpha_1^* \lambda^* \delta - q\mu_1^2}{\mu_1}, \tag{27}$$

$$K_2 < \frac{(1 - \alpha_1^*) \lambda^* \delta - q\mu_2^2}{\mu_2}. \tag{28}$$

We assume that $\lambda^*$ and $\alpha_1^*$ are obtained in the previous stages and $\delta$, $c$, $p$ are fixed, and we have proceed numerically in the analysis of Stage I, as regards the BRs and the Nash equilibria.

## 4. Results and Discussion

In this section, we will present some numerical results obtained with the above model. First, we explore the strategic decision, concerning the MNOs network capacity $(\mu_1^*, \mu_2^*)$, MVNO traffic split factor $\alpha_1^*$, packets mean arrival rate $\lambda^*$ and MNOs Profits $(\Pi_1^*, \Pi_2^*)$. Second, we will compare the multi-MNO MVNO models and the single-MNO. Please note that the numerical computations performed are a numerical solving of the system of Equations (23) and (24) to compute the Nash equilibria. For this purpose, we set $K_1 = K_2$ and $K_1 < K_2$. We have conducted a series of numerical experiments to obtain a better understanding of the scenario from the economic interactions. The values for the parameters, if not stated otherwise, are the ones shown in Tables 2–4 and Table A1 and explained below:

- $K_i$ is a unit cost of acquisition [60] and the values assigned to this parameter in analysis satisfy the restrictions (27) and (28).
- $q$ is the adjustment cost parameter [60] and values assigned to this parameter in analysis satisfy restrictions $q < K_i$ and $q > 0$.
- $p$ is the price charged by MNO and value assigned is greater than MNOs fee for obtaining positive MVNO profits in Equation (9).
- $\delta$ is the fee paid by the MVNO and value assigned is $\delta < p$.
- $c$ is the conversion factor to monetary units, and the value assigned is 2 . This parameter has not further relevance in the numerical computations.

### 4.1. MNOs Investment Costs with $K_1 = K_2$

The results of this section were obtained based on the parameter values shown in Table 2. This parameter setting illustrates a basic symmetric scenario between two MNOs of our model.

**Table 2.** Parameter values for numerical computations.

| Parameter | Value |
|:---:|:---:|
| $K_1$ | $[0, 1]$ |
| $K_2$ | $[0, 1]$ |
| $q$ | $\{0.015, 0.025, 0.5\}$ |
| $p$ | 1.8 |
| $\delta$ | 1.6 |
| $c$ | 2 |

Figure 3 shows MNOs network capacity $(\mu_1^*, \mu_2^*)$ as a function for $K_i$ $(K_1 = K_2)$ with different values of $q$. When $K_i$ and $q$ increase, MNOs network capacity decreases, as shown in Figure 3a,b. In both the above cases, the behaviour MNOs network capacity is the same; i.e., MNO 1 network capacity is equal to MNO 2 network capacity $(\mu_1^* = \mu_2^*)$.

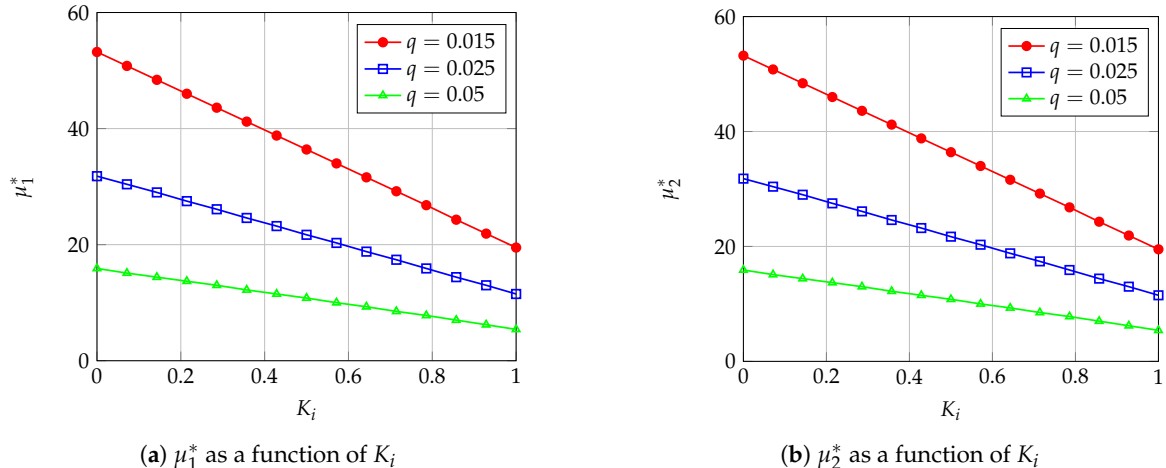

(**a**) $\mu_1^*$ as a function of $K_i$    (**b**) $\mu_2^*$ as a function of $K_i$

**Figure 3.** MNOs network capacity as a function of $K_i$.

Figure 4 shows $\alpha_1^*$ and $\lambda^*$ as a function for $K_i$ with different values of $q$. The impact on $\alpha_1^*$ when $K_i$ increases from 0 to 1 with different values of $q$ is negligible, as shown in Figure 4a, $\alpha_1^*$ stays constant and equals $\frac{1}{2}$. The described behaviour is consistent with the analysis performed in Proposition 2 if $\mu_1^* = \mu_2^*$ then $\alpha_1 = \frac{1}{2}$. $\lambda^*$ decreases when $K_i$ increases from 0 to 1 and $q$ also increases as shown in Figure 3b. The described behaviour is consistent with the analysis performed in Proposition 1 for $\alpha_1 \in (0,1)$, the values that $\lambda$ can take are defined by Equation (12).

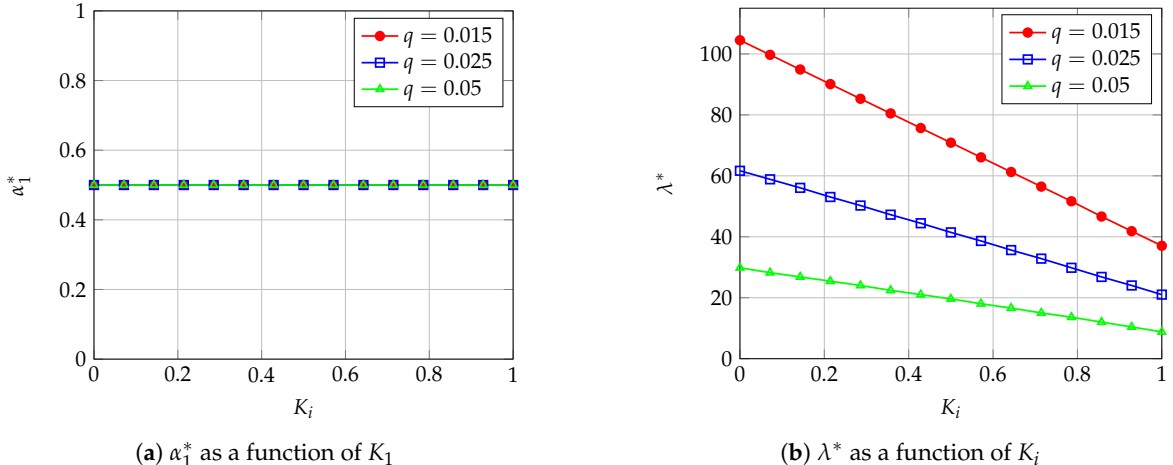

(**a**) $\alpha_1^*$ as a function of $K_1$    (**b**) $\lambda^*$ as a function of $K_i$

**Figure 4.** $\alpha_1^*$ and $\lambda^*$ as a function of $K_i$.

Figure 5 shows MNOs profits as a function of $K_i$ with different values of $q$. MNOs profits decrease when MNOs investment costs increase as shown in Figure 5a,b. When MNOs investment costs are equals $(K_1\mu_1 + q\mu_1^2 = K_2\mu_2 + q\mu_2^2)$ MNOs profits will be equal $(\Pi_1 = \Pi_2)$ and its behaviour is the one described above when investment costs increase. Likewise, MNO 2 does not subtract network traffic from MNO 1 but increases the global traffic of the system $(\lambda)$, that is, $\lambda = \alpha_1\lambda_1 + (1 - \alpha_1)\lambda_2$.

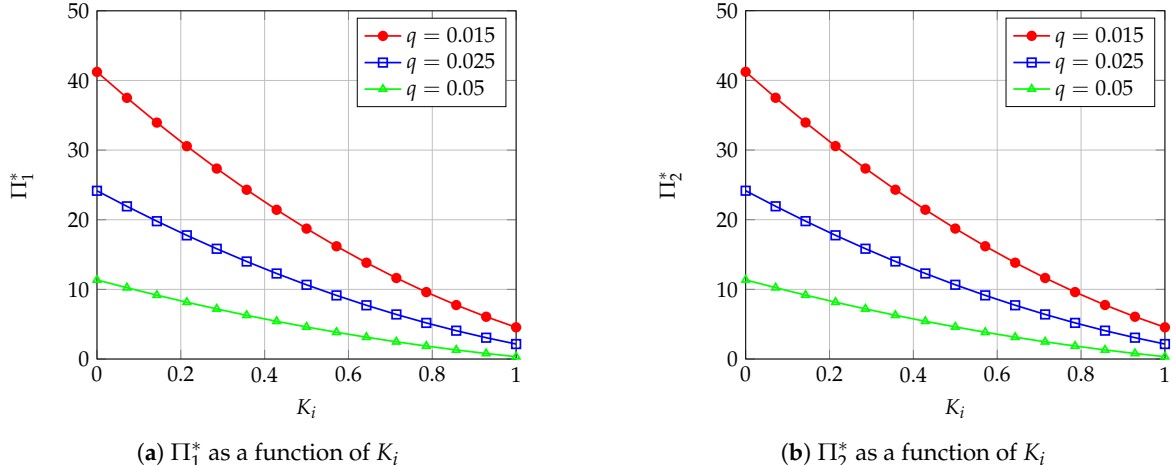

(**a**) $\Pi_1^*$ as a function of $K_i$        (**b**) $\Pi_2^*$ as a function of $K_i$

**Figure 5.** MNOs profits as a function of $K_i$ .

The results show that increase or decrease MNOs investment costs have an impact on MNOs network capacity (e.g., $\mu_1^*$ in Figure 3a). When MNOs investment costs are equal ($K_1\mu_1 + q\mu_1^2 = K_2\mu_2 + q\mu_2^2$) the traffic split factor will be $\frac{1}{2}$; i.e., the MVNO splits its network traffic equally among MNOs. Consequently, when MNOs investment costs of both operators increase MNOs network capacity decreases and this also causes MVNO users' mean arrival rate to decrease. The described behaviour impact on MNOs profits, i.e., each MNO will adopt a strategy that will always provide greater utility to one player, regardless of the strategy of the other MNO (Dominant Strategy).

*4.2. MNOs Investment Costs with $K_1 \neq K_2$*

The results of this section were obtained based on the parameter values shown in Table 3. This parameter setting illustrates a scenario where the MNOs investment costs are different.

**Table 3.** Parameter values for the numerical computations.

| Parameter | Value |
|:---:|:---:|
| $K_1$ | $[0, 1]$ |
| $K_2$ | $0.45$ |
| $q$ | $\{0.015, 0.025, 0.05\}$ |
| $p$ | $1.8$ |
| $\delta$ | $1.6$ |
| $c$ | $2$ |

Figure 6 shows the MNOs network capacity ($\mu_1^*, \mu_2^*$) as a function of $K_1$ for different values of $q$ and $K_2 = 0.45$. MNO 1 network capacity ($\mu_1^*$) decreases when $K_1$ increases from 0 to 1 for different values of q as shown in Figure 6a. At the same time, when $K_1$ and $q$ increase, MNO 2 network capacity stays constant as shown in Figure 6b.

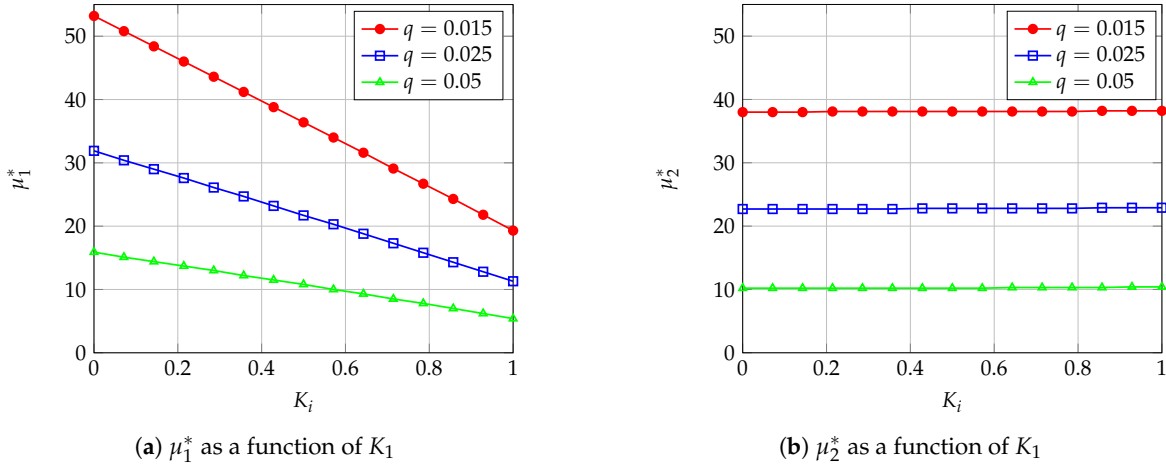

**(a)** $\mu_1^*$ as a function of $K_1$

**(b)** $\mu_2^*$ as a function of $K_1$

**Figure 6.** MNOs network capacity as a function of $K_1$ with $K_2 = 0.45$.

Figure 7 shows $\alpha_1^*$ and $\lambda^*$ as a function of $K_1$ with different values of $q$ and $K_2 = 0.45$. When $K_1$ increases from 0 to $K_1 < K_2$ and $q > 0$, $\alpha_1^*$ is greater than $\frac{1}{2}$, i.e., MNO 2 investment cost is greater than MNO 1 investment cost. The MNO 2 network capacity is lower than MNO 1 network capacity as shown in Figure 6. When $K_1 = K_2$ and $q > 0$, $\alpha_1^*$ stays constant and it is equal to $\frac{1}{2}$. When $K_1$ increases from $K_1 > K_2$ to 15 and $q > 0$, $\alpha_1^*$ is lower than $\frac{1}{2}$, i.e., MNO 2 investment cost is lower than MNO 1 investment cost and the MNO 1 network capacity is lower than MNO 2 network capacity as shown in Figure 6. When $K_1$ increases from 0 to 1 and $q > 0$, $\lambda$ decreases as shown in Figure 7b; i.e., the MVNO users mean arrival rate depends on MNOs network capacity. The described behaviour is consistent with the analysis performed in Proposition 1 for $\alpha_1 \in (0,1)$; the values that $\lambda$ can take are defined by Equation (12).

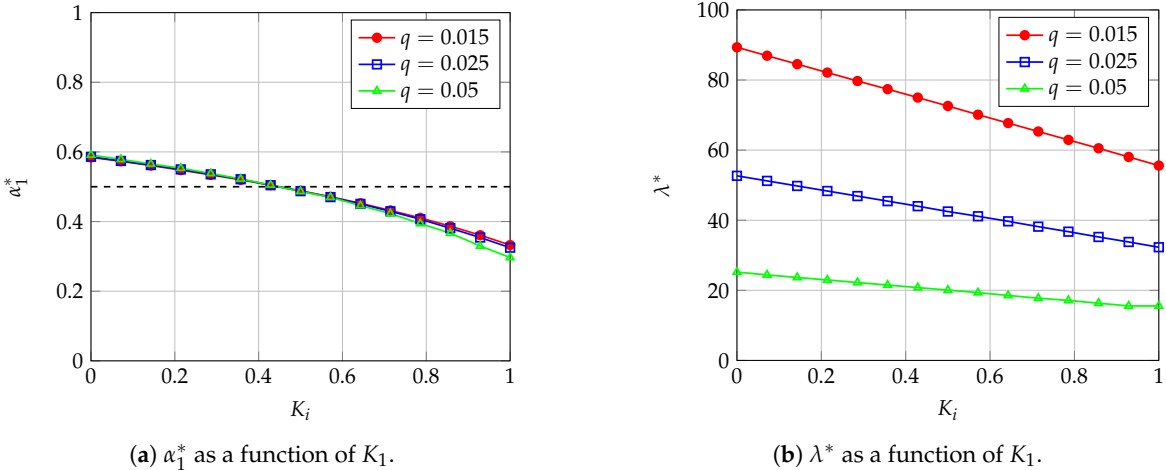

**(a)** $\alpha_1^*$ as a function of $K_1$.

**(b)** $\lambda^*$ as a function of $K_1$.

**Figure 7.** $\alpha_1^*$ and $\lambda^*$ as a function of $K_1$ with $K_2 = 0.45$.

Figure 8 shows MNOs profits as a function of $K_1$ and different values of $q$ and $K_2 = 0.45$. MNO 1 profits decrease when $K_1$ increases from 0 to 1; this is due to the decrease in network capacity offered by MNO 1. This effect is most noticeable when $q$ ($q\mu_i^{*2}$) increases, as shown in Figure 8a. On the other hand, the impact of $K_1$ parameter of MNO 1 investment costs on the MNO 2 profits is negligible as shown in Figure 8b. However, MNO 2 profits decrease when $q$ increases even when MVNO assigns more network traffic to the MNO 2, i.e., $\alpha_2 > \alpha_1$ ($1 - \alpha_1 > \alpha_1$), as shown in Figure 7a. This occurs when MNOs investment costs increase and MNO 2 network capacity $\mu_2^*$ decreases as shown in Figure 6b.

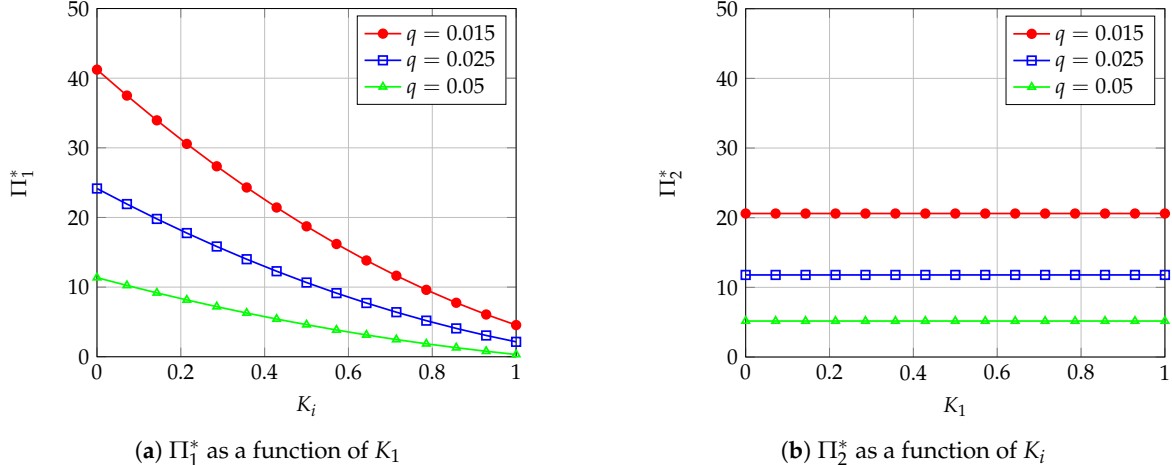

(**a**) $\Pi_1^*$ as a function of $K_1$  (**b**) $\Pi_2^*$ as a function of $K_i$

**Figure 8.** MNOs profits as a function of $K_1$ and $K_2 = 0.45$.

We can conclude that when $\alpha_1^* > \frac{1}{2}$ ($K_1 < K_2$) MNO 1 network capacity is greater than MNO 2 network capacity and therefore MNO 1 profits will be higher. In the same way, when $\alpha_1^* = \frac{1}{2}$ ($K_1 = K_2$) MNO 1 network capacity is equal to MNO 2 network capacity and therefore MNOs profits will be equal. When $\alpha_1^* < \frac{1}{2}$ ($K_1 > K_2$), MNO 2 network capacity is greater than MNO 1 network capacity and therefore MNO 2 profits will be higher, i.e., each MNO will adopt a strategy that will always provide greater utility to one player, regardless of the strategy of the other MNO (Dominant Strategy).

Note that, the behaviour when we set the parameters $K_2$ from 0 to 1 and $K_1 = 0.45$, $p = 0.08$, $\delta = 0.6$ and $c = 1$ is symmetrical; i.e., $\mu_i^*$, $\lambda^*$, $\Pi_i$ exhibits the same behaviour as the one described above but only when $K_2$ increases from 0 to 1, as shown in Figure A4 in Appendix B.

### 4.3. Comparison between Multi-MNO MVNO and Single-MNO Models

Finally, this section discusses the multi-MNO MVNO models and single-MNO, that is, the comparison among the MNOs network capacity, MVNO users mean arrival rate, MNOs profits and MVNO profits obtained in Section 4.2 with those obtained in Appendix A. For this purpose, the parameters' values are displayed in Table 4.

**Table 4.** Parameter values for the numerical computations.

| Parameter | Value |
|:---------:|:-----:|
| $K_0$ | $[0, 1]$ |
| $K_1$ | $[0, 1]$ |
| $K_2$ | 0.45 |
| $q$ | 0.025 |
| $p$ | 1.8 |
| $\delta$ | 1.6 |
| $c$ | 2 |

Figure 9 shows the MNOs network capacity (multi-MNO) and MNO 0 (single-MNO) network capacity as a function of $K_1$ ($K_0 = K_1$) with $q = 0.025$ and $K_2 = 0.45$. Firstly, the MNO 1 network capacity is higher than the MNO 0 network capacity. Secondly, the impact on network capacity (MNO 0, MNO 1) when $K_1$ increases from 0 to 1 is notorious as shown in Figure 9, when $K_1$ increases the network capacity (MNO 1, MNO 0) decreases. Thirdly, MNO 2 network capacity stays constant when $K_1$ increases from 0 to 1. In addition, MNO 2 network capacity is lower than MNO 1 network capacity when $K_1 < K_2$ ($K_2 = 0.45$) and MNO 2 network capacity is greater than MNO 1 network capacity. This behaviour is described in Section 4.2.

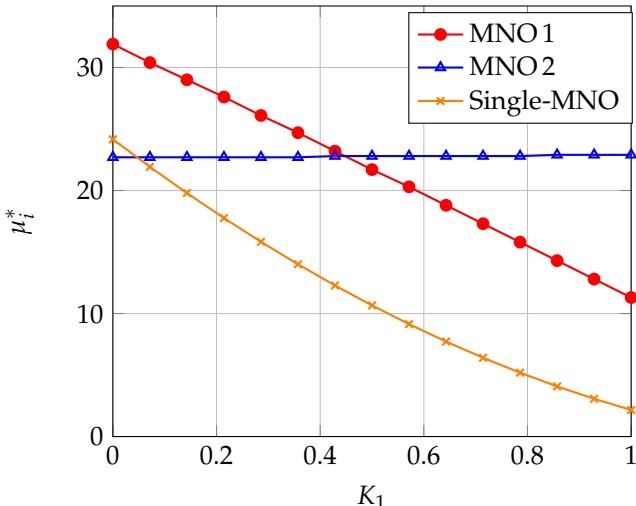

**Figure 9.** MNO $i$'s network capacity as a function of $K_1$ with $K_2 = 0.45$.

Figure 10 shows $\lambda^*$ as a function of $K_1$ ($K_0 = K_1$) with $q = 0.025$ and $K_2 = 0.45$. The impact on $\lambda_0$ (single-MNO) and $\lambda$ (multi-MNO MVNO) when $K_1$ increases from 0 to 1 it is high, due to users mean arrival ($\lambda_0, \lambda$) decreases. Besides, the users mean arrival when MVNO shares its network traffic with two MNOs, it is higher than when it only shares with one MNO.

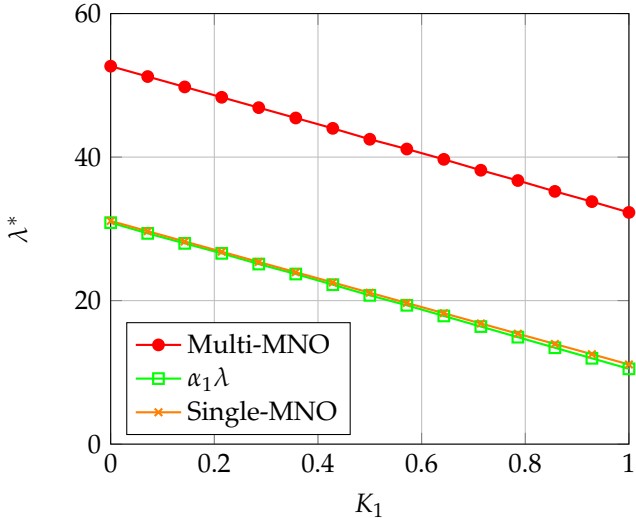

**Figure 10.** MNOs and MVNO network capacity as a function of $K_1$ with $K_2 = 0.45$.

Figure 11 shows the equilibrium profits of each operator ($\Pi_1, \Pi_2, \Pi_0$) as a function of $K_1$ ($K_0 = K_1$) with $q = 0.025$ and $K_2 = 0.45$. We observe that when MNO 0 and MNO 1 profits are equal when $K = K_1$ increases from 0 to 1 the MNO 1 profits decreases. At the same time, MNO 2 profits remain constant given that for this we consider $K_2 = 0.45$, i.e., the MNO 2 investment costs remain constants. Besides, with $K_0 = K_2$ and $K_1 = 0.45$ exhibit the same behaviour as the one described above but MNO 2 and MNO 0 profits are equal and decrease when $K_2$ increases, as shown in Figure 11b.

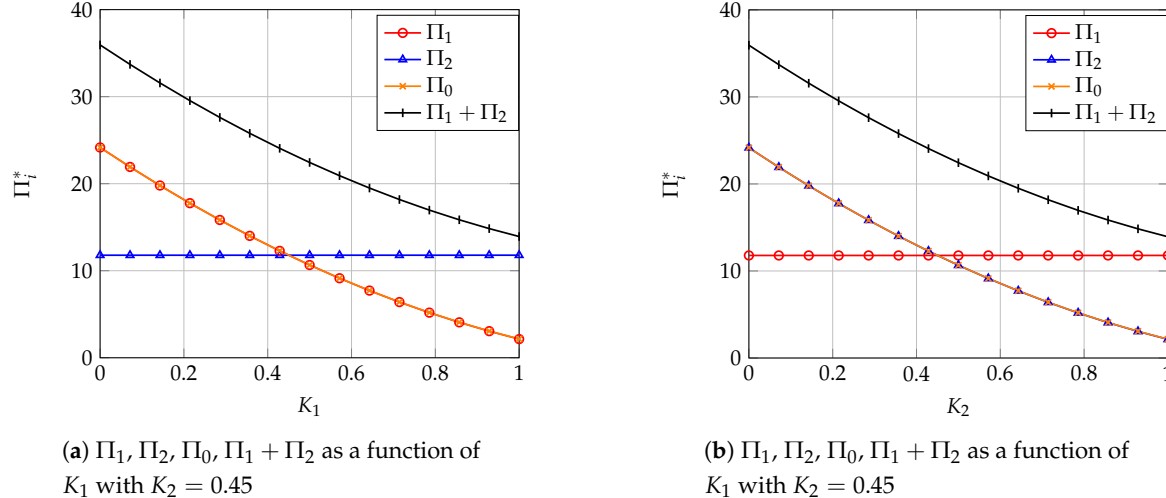

(**a**) $\Pi_1$, $\Pi_2$, $\Pi_0$, $\Pi_1 + \Pi_2$ as a function of $K_1$ with $K_2 = 0.45$

(**b**) $\Pi_1$, $\Pi_2$, $\Pi_0$, $\Pi_1 + \Pi_2$ as a function of $K_1$ with $K_2 = 0.45$

**Figure 11.** MNOs equilibrium profits.

When the profit that the MNO would get in a single-MNO model is compared, $\Pi_0$ (the Appendix A), we conclude that the MNO profits are the same for both of the analyzed models. However, the investment costs in the single-MNO model are lower to obtain the same profits as in the multi-MNO MVNO model, given that MNO 0 network capacity is less than the MNO 1 network capacity required in model multi-MNO MVNO. Besides, we have determined a profits lump sum, which includes MNOs profits on a multi-MNO MVNO model as shown in Figure 11. The global profits $\Pi_1 + \Pi_2$ have the same behaviour described above for the MNOs profits when $K_1$ decreases as shown in Figure 11a,b, i.e., also decrease when investment costs increase. The global profits of a multi-MNO MVNO model is greater than a single-MNO model.

Figure 12 shows the MVNO profits of multi-MNO MVNO and Single-MVNO models as a function of $K_1$ with $q = 0.025$ and $K_2 = 0.45$. We observe that when MVNO has an agreement with two MNOs, its profits are greater than when the MVNO has an agreement with only one MNO. The explanation for these results is related to the fact that $\lambda > \lambda_0$ as shown in Figure 10. Besides, we observe that the MVNO suffers a reduction in its profit in both cases. This is due to the fact that the MNOs network capacity ($\mu_1, \mu_2$) decreases when investment cost increases.

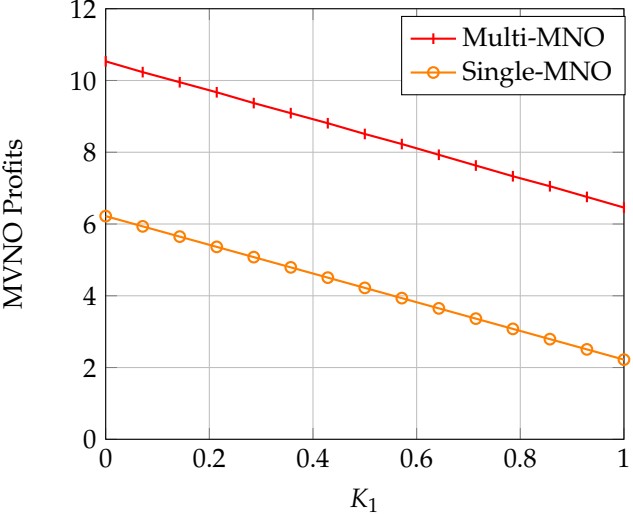

**Figure 12.** MVNO equilibrium profits as a function of $K$ with $K_2 = 0.45$.

The results show that the users mean arrival in multi-MNO MVNO model is larger than the single-MNO model ($\lambda > \lambda_0$); i.e., the MVNO can subscribe a higher number of users as shown in Figure 10, due to the fact the entry of the MNO 2 increases the MVNO network capacity ($\lambda = \alpha_1\lambda + (1 - \alpha_1)\lambda$). Consequently, MVNO profits in the multi-MNO MVNO model are larger than single-MNO model as shown in Figure 12. Likewise, with regard to MNOs, the network capacity $\mu_1$ in multi-MNO MVNO model is greater than $\mu_0$ in the single-MNO model as shown in Figure 9. However, the MNO 0 and MNO 1 profits are equal and decrease when $K_1$ increases as shown in Figure 11; i.e., MNOs will need more network capacity in the multi-MNO MVNO model to get the same profits as in the single-MNO model.

## 5. Conclusions

In this paper, a novel business model for multi-MNO and MVNO relationship has been studied. We have studied the feasibility of the model from a positive-profit point of view for all the actors of the multi-MNO MVNO model and we compared the results with a single-MNO model. Our main results suggest that in one business model, named multi-MNO MVNO, the MVNO provides service to a greater number of users than the single-MNO model. As a consequence, the multi-MNO MVNO model generates more revenue to MVNO than the Single-MVNO Model. In this context, MNO 1 profits are equal to the MNO 0 profits because each MNO will adopt a strategy that will always provide greater utility to one player, regardless used by the strategy of the other MNO (Dominant Strategy). In the multi-MNO MVNO model, the MNO 1 network capacity must be greater to obtain the same benefits as in the single-MNO model. However, the entry of the second operator (MNO 2) into the system is desirable not only from the point of view of the resource usage efficiency but also from the users' point of view, but also from the operators. The above is explained by the entry of MNO 2 into the market which allows to increase the number of users to MVNO and generate positive profits for all operators in the multi-MNO MVNO model. Moreover, the new operator MNO 2 does not subtract users from the incumbent operator (MNO 1) but rather increases the capacity of the MVNO, which will distribute its network traffic according to $\alpha_1$. Additionally, we have shown that multi-MNO MVNO model is feasible, since it provides incentives to operators compared with the single-MNO scenario. For each parameter configuration, there is a range of values of $K_i$ and $q$, for which a lump sum profit can be designed so that MNO 1 has an incentive to let MNO 2 enter.

Given that the three stages have been shown feasible under specific conditions of users' subscription decision, from the optimal division of the MVNO network traffic and the positive-profits operators, we can conclude that the whole multi-MNO MVNO model is conditionally feasible from an economic point of view.

**Author Contributions:** Conceptualization, L.G.; Validation, P.M. and L.G.; Formal analysis, E.J.S.C., L.G. and P.M.; Investigation, E.J.S.C.; Visualization, E.J.S.C.; Writing-original draft, E.J.S.C.; Writing-review and editing, L.G. and P.M. All authors have read and agreed to the published version of the manuscript.

**Funding:** This work has been supported by the Spanish Ministry of Science, Innovation and Universities (MCIU/AEI) and the European Union (FEDER/UE) through Grant PGC2018-094151-B-I00 and partially supported by Politécnica Salesiana University (Salesian Polytechnic University) in Ecuador through a Ph.D. scholarship granted to the first author.

**Conflicts of Interest:** The authors declare no conflict of interest.

## Appendix A. Single-MNO

In this appendix, we study the case where only the network operator provides the service to MVNO, i.e., this can occur when the MVNO unilaterally decides to send all network traffic to a single MNO, as shown in Figure A1. The first stage described in Section 3 is reduced to an optimal decision by MNO. The second stage in this case where $\alpha_1$ becomes 0 for MNO 2 and 1 for MNO 1 respectively, as decided by MVNO. The third stage is reduced to the choice between the service

provided by the MVNO or no service−the utility is then assumed to be zero. Therefore, we will call this case single-MNO.

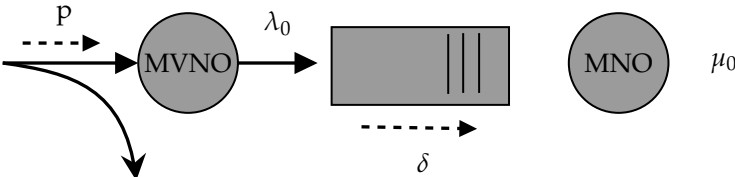

**Figure A1.** System Model.

Thus, from Equation (10) the revenue for a MNO

$$\Pi_0 = \lambda_0 \delta - K_0 \mu - q_0 \mu^2. \tag{A1}$$

- MVNO Service Provision

  The average system delay $T$ from Equation (4)

  $$T = \frac{1}{\mu - \lambda_0}. \tag{A2}$$

  Thus, the users utility in this case is:

  $$U_{m_0} = c\, T^{-1} - p. \tag{A3}$$

  where $T = \left(\frac{1}{\mu - \lambda_0}\right)$, and substituting $T$ into Equation (A3), we obtain

  $$U_{m_0} = c\left(\frac{1}{\mu - \lambda_0}\right)^{-1} - p. \tag{A4}$$

  Analyzing the users' subscription decision, we observe that given a price $p$ announced by the operator, the Wardrop equilibrium will be as follows.

  – Case I: The number of users subscribing increases until the utility is zero. Therefore, the condition for this case is
  $$U_{m_0} = 0. \tag{A5}$$

    Solving Equation (A4) under the Condition (A5), the number of users is then

    $$\lambda_0 = \left(\mu - \frac{p}{c}\right). \tag{A6}$$

  – Case II: The price in Equation (A4) is so high that the utility is always negative. Therefore the condition for this case is

    $$U_{m_0} < 0. \tag{A7}$$

    Under the Condition (A7), the users do not subscribe the service. Therefore, the number of users is

    $$\lambda_0 = 0. \tag{A8}$$

Assuming equilibrium in Case I, we can obtain the MVNO profits $\Pi_{m_0}^*$ substituting $\lambda_0^*$ into Equation (9).

$$\Pi_{m0}^* = (p - \delta)\left(\mu - \frac{p}{c}\right). \tag{A9}$$

- MNO Profits

At this point, we proceed to analyze the network capacity $\mu^*$ given by the value of $\lambda^*$ from the previous section. We can compute the profit for MNO in the monopoly scenario substituting Equation (A6) into Equation (A1)

$$\Pi_0(\mu) = \mu(\delta - K_0 - \mu q_0) - \frac{\delta p}{c}. \tag{A10}$$

We can maximize the profit by setting its derivative with respect to the price ($\mu_i$) equal to zero, the result of ($\mu_i^*$) is

$$\mu_0^* = \frac{\delta - K_0}{2q_0}. \tag{A11}$$

Finally, we can obtain the maximum profit substituting Equation (A11) into Equation (A1)

$$\Pi_0^* = \frac{c(K - \delta)^2 - 4\delta pq}{4cq}. \tag{A12}$$

## Appendix B. MNOs Profits with $K_1 = 0.45$ and $0 < K_2 < 1$

The results displayed in this section have been obtained considering the values of the parameters displayed in Table A1. This parameter setting illustrates a scenario where the MNOs investment costs are different.

**Table A1.** Parameter values for the numerical computations.

| Parameter | Value |
|-----------|-------|
| $K_1$ | 0.45 |
| $K_2$ | $[0, 1]$ |
| $q$ | $\{0.015, 0.025, 0.5\}$ |
| $p$ | 1.8 |
| $\delta$ | 1.6 |
| $c$ | 2 |

Figure A2 shows MNOs network capacity ($\mu_1^*, \mu_2^*$) as a function of $K_2$ for different values of $q$ and $K_1 = 0.45$. When $K_2$ and $q$ increase, MNO 1 network capacity stays constant as shown in Figure A2a. At the same time, MNO 2 network capacity ($\mu_2^*$) decreases when $K_2$ increases from 0 to 1 for different values of q as shown in Figure A2b.

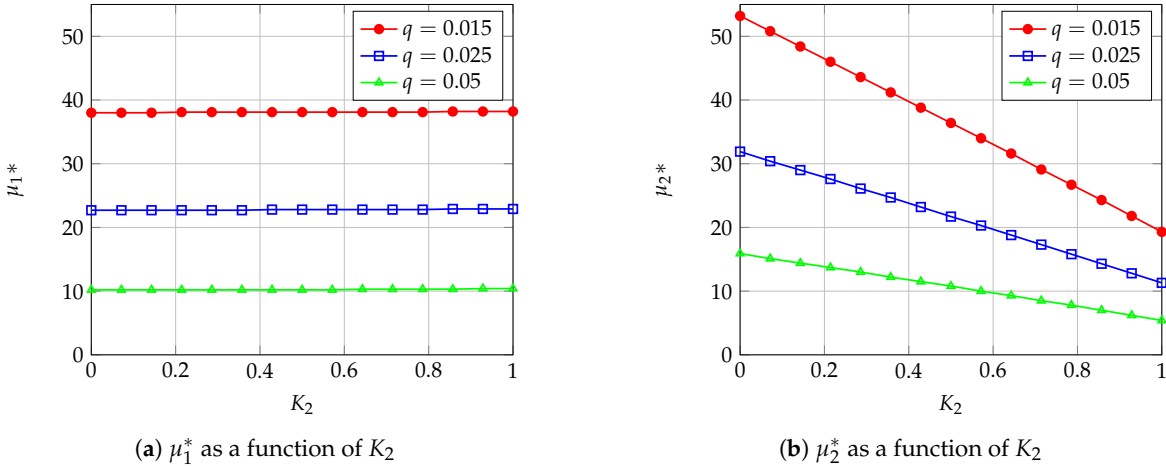

(a) $\mu_1^*$ as a function of $K_2$        (b) $\mu_2^*$ as a function of $K_2$

**Figure A2.** MNOs network capacity as a function of $K_2$ with $K_1 = 0.45$.

Figure A3 shows $\alpha_1^*$ and $\lambda^*$ as a function of $K_2$ with different values of $q$ and $K_1 = 0.45$. When $K_2$ increases from 0 to $K_2 < K_1$ and $q > 0$, $\alpha_1^*$ is lower than $\frac{1}{2}$, i.e., MNO 2 investment cost are lower than MNO 1 investment cost. The MNO 1 network capacity is lower than MNO 2 network capacity as shown in Figure A2. When $K_2 = K_1$ and $q > 0$, $\alpha_1^*$ stays constant and $\alpha_1^*$ is equal $\frac{1}{2}$. In addition, when $K_2$ increases from $K_2 > K_1$ to 1 and $q > 0$, $\alpha_1^*$ is greater than $\frac{1}{2}$, i.e., MNO 2 investment cost are greater than MNO 1 investment cost and the MNO 2 network capacity is lower than MNO 1 network capacity as shown in Figure A2. When $K_2$ increases from 0 to 1 and $q > 0$, $\lambda$ decreases as shown in Figure A3b; i.e., the MVNO users mean arrival rate depends on the MNOs network capacity. The described behaviour is consistent with the analysis performed in Proposition 1 for $\alpha_1 \in (0, 1)$, the values that $\lambda$ can take are defined by Equation (12).

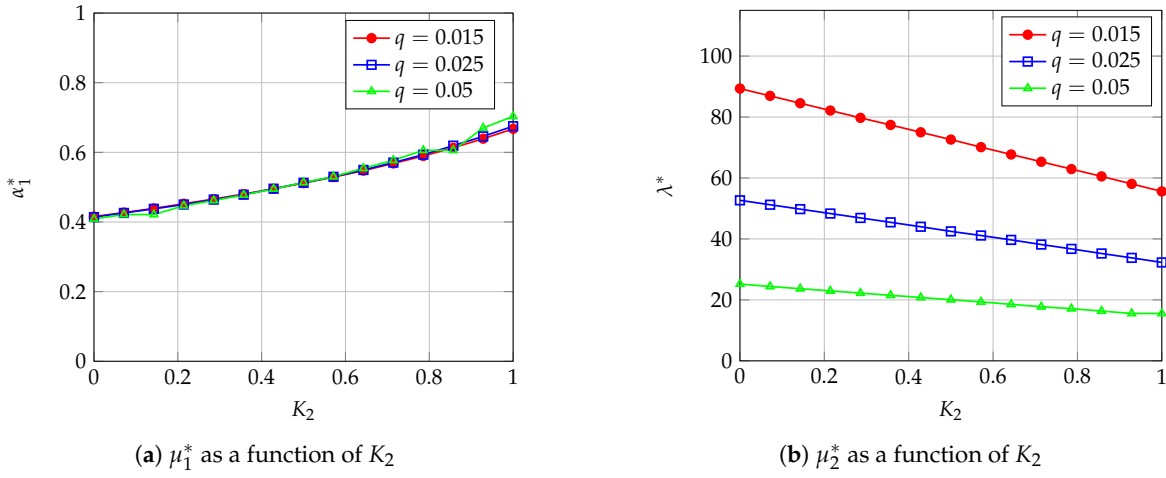

(a) $\mu_1^*$ as a function of $K_2$        (b) $\mu_2^*$ as a function of $K_2$

**Figure A3.** $\alpha_1^*$ and $\lambda^*$ a function of $K_2$ with $K_1 = 0.45$.

Figure A4 shows MNOs profits as a function of $K_2$ and different values of $q$ and $K_1 = 0.45$. MNO 2 profits decrease when $K_2$ increases from 0 to 1; this is due to the decrease in network capacity offered by MNO 2. This effect is most noticeable when $q$ ($q\mu_i^{*2}$) increases, as shown in Figure A4b. On the other hand, the impact of $K_2$ of MNO 2 investment costs on MNO 1 profits is negligible as shown in Figure A4a. However, MNO 1 profits decrease when $q$ increases even if MVNO assigns more network traffic to MNO 1, i.e., $\alpha_1 > \alpha_2$ ($\alpha_1 > 1 - \alpha_1$), as shown in Figure A3a. This occurs when MNOs investment costs increase and MNO 1 network capacity $\mu_1^*$ decreases as shown in Figure A2a.

This arises when MNO investment cost increase and MNO 1 network capacity $\mu_1^*$ decreases, as related in Section 4.1.

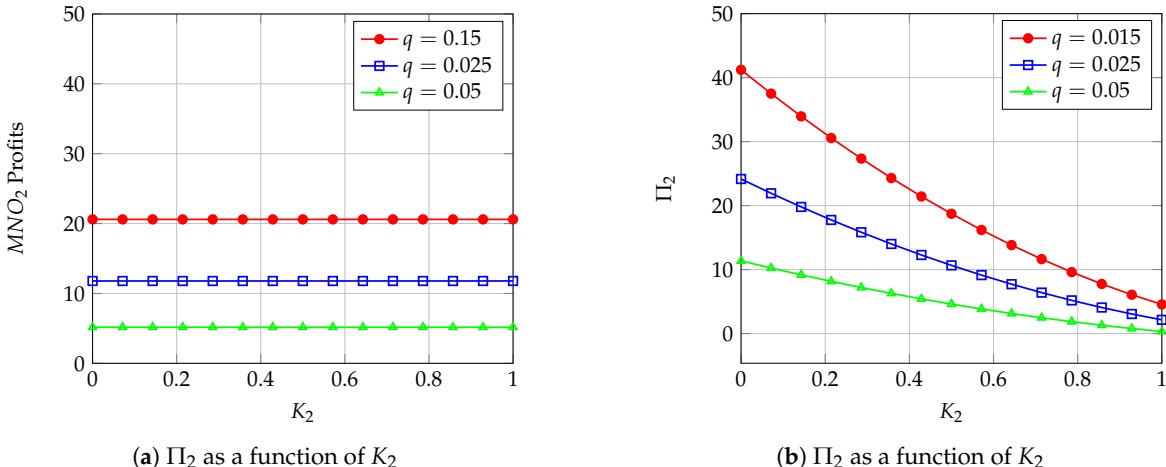

(a) $\Pi_2$ as a function of $K_2$      (b) $\Pi_2$ as a function of $K_2$

**Figure A4.** MNOs profits as a function of $K_2$ with different values of $q$ and $K_1 = 0.1$, $p = 0.08$, $c = 1$.

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
