# Peer review of "Game Theoretical Analysis of a Multi-MNO MVNO Business Model in 5G Networks"

_electronics, doi:10.3390/electronics9060933_

Round 1

Reviewer 1 Report

Dear Authors

I thank you for the thorough answers to my comments to the paper. I think the authors have addressed my comments in a reflective manner and done appropriate modifications to the paper. 

Author Response

Dear Reviewer,

We have received the results of the revision of the manuscript number electronics 779934 entitled Game theoretical analysis of a multi-MNO MVNO Business Model in 5G Networks, which was submitted to Electronics.

We truly appreciate the comments, which we have incorporated in the revised version. Thank you very much for your effort. We send a reply to your comments in the file attached.

We have incorporated all the above arguments and clarifications in the revised manuscript.

Thus, we hereby resubmit the manuscript with the conviction that it will meet the standards expected by the Electronics editorial board and readership.

Yours faithfully,

The authors.

Reviewer 2 Report

The authors have carefully revised the draft and I suggest to accept this paper after revising wording and formatting.

Author Response

(The authors gave the same response as above.)

Reviewer 3 Report

The article analyses the queue theory for resource optimization. The analysis is in clear and detail however from the reviewer's side there are some issues as:

  • There is no novel with Queue theory with M/M/1. The article is just one use case.
  • There is no connection to 5G. The analysis is for general application not only for 5G. Actually, in 3G and 4G this business model is already developed.
  • Commonly, end user is total blind for network selection in the point of  resource utilization. 

Author Response

(The authors gave the same response as above.)

Round 2

Reviewer 3 Report

Thanks for author contribution.

This manuscript is a resubmission of an earlier submission. The following is a list of the peer review reports and author responses from that submission.

Round 1

Reviewer 1 Report

Dear Authors.

The main criticism of this paper is its underlying idea. The interaction between the MVNO and the MNO is not realistic, which in turn makes the whole model unrealistic. The method used is fine, even though equilibrium analysis is not among the most modern approaches to analyze digital markets. The amount of results is reasonable. 

Reviewer 2 Report

In this paper, the authors propose a business model to be implemented by two MNOs and an MVNO and analyze the proposal using a strategic game. The research work presented in this paper looks solid and interesting. Some suggestions are presented as follows.

  1. The authors are suggested to highlight the main difference between the published works and the proposed work in this paper before the details.
  2. The authors are suggested to explain more about the value of simulation parameters presented in page 11.

Reviewer 3 Report

The article discusses the economic feasibility of a business model for multi-Mobile Network Operators (MNOs) and Mobile Virtual Network Operators (MVNO) using queue analysis. In the point of the reviewer, there are some following discussions as:

  • The queue analysis is not novel. The article is an analysis of the business issues using queue theory.
  • In a practical scenario, the number of users is limited. The increasing of MVNO based on the MNO is just one sharing of users. The increasing of the user usually bases on the new providing service. The business analysis which is considered in the article is for resource utilization. It does not cover the mentioned issue.
  • The analysis is clear however it is novel work.